# Advanced Molecular Characterisation in Relapsed and Refractory Paediatric Acute Leukaemia, the Key for Personalised Medicine

**DOI:** 10.3390/jpm12060881

**Published:** 2022-05-27

**Authors:** Víctor Galán-Gómez, Nerea Matamala, Beatriz Ruz-Caracuel, Paula Valle-Simón, Bárbara Ochoa-Fernández, Pilar Guerra-García, Alicia Pernas-Sánchez, Jordi Minguillón, Berta González, Isabel Martínez-Romera, Sonsoles San Román-Pacheco, Pablo Estival-Monteliú, Adrián Ibáñez-Navarro, Antonio Pérez-Martínez, Adela Escudero-López

**Affiliations:** 1Paediatric Haemato-Oncology Department, La Paz University Hospital, 28046 Madrid, Spain; victor.galan@salud.madrid.org (V.G.-G.); barbaraochoaf@gmail.com (B.O.-F.); pilar.guerra@salud.madrid.org (P.G.-G.); berta.gonzalez@salud.madrid.org (B.G.); imromera@salud.madrid.org (I.M.-R.); sonsoles.sanroman@salud.madrid.org (S.S.R.-P.); aperezmartinez@salud.madrid.org (A.P.-M.); 2Genetics Department (INGEMM), La Paz University Hospital, 28046 Madrid, Spain; nmatamala@isciii.es (N.M.); ingemm.bruz@gmail.com (B.R.-C.); alicia.pernas@salud.madrid.org (A.P.-S.); jordi.minguillon@gmail.com (J.M.); 3Clinical Pharmacology Department, La Paz University Hospital, 28046 Madrid, Spain; paula.ucicec@gmail.com; 4School of Medicine, Autonomous University of Madrid, 28046 Madrid, Spain; pabloestival@gmail.com (P.E.-M.); adrianibnavarro@gmail.com (A.I.-N.)

**Keywords:** paediatric acute leukaemia, refractory, relapsed, NGS, personalised medicine, advanced molecular characterisation, genetics

## Abstract

Relapsed and refractory (R/r) disease in paediatric acute leukaemia remains the first reason for treatment failure. Advances in molecular characterisation can ameliorate the identification of genetic biomarkers treatment strategies for this disease, especially in high-risk patients. The purpose of this study was to analyse a cohort of R/r children diagnosed with acute lymphoblastic (ALL) or myeloid (AML) leukaemia in order to offer them a targeted treatment if available. Advanced molecular characterisation of 26 patients diagnosed with R/r disease was performed using NGS, MLPA, and RT-qPCR. The clinical relevance of the identified alterations was discussed in a multidisciplinary molecular tumour board (MTB). A total of 18 (69.2%) patients were diagnosed with B-ALL, 4 (15.4%) with T-ALL, 3 (11.5%) with AML and 1 patient (3.8%) with a mixed-phenotype acute leukaemia (MPL). Most of the patients had relapsed disease (88%) at the time of sample collection. A total of 17 patients (65.4%) were found to be carriers of a druggable molecular alteration, 8 of whom (47%) received targeted therapy, 7 (87.5%) of them in addition to hematopoietic stem cell transplantation (HSCT). Treatment response and disease control were achieved in 4 patients (50%). In conclusion, advanced molecular characterisation and MTB can improve treatment and outcome in paediatric R/r acute leukaemias.

## 1. Introduction

Paediatric cancer is the main cause of disease-related mortality in children in developed countries [1]. Haematological malignancies are the most frequent cancer in this population, and within them, acute lymphoblastic leukaemia (ALL) is the predominant type, comprising 20% of total cancer cases occurring before the age of 19 [2,3].

The survival rates of paediatric leukaemia have dramatically increased over the last few decades; the current 5-year overall survival for ALL is 90% [4], constituting 65–70% for AML [5]. This achievement is due, among other reasons, to the establishment of international collaborative research platforms, the improvement of risk stratification and the introduction of multimodal innovative treatments, such as immunotherapy or chimeric antigen receptor (CAR) -T cell therapy [6,7].

Despite such improvements, R/r patients still represent a high-risk population with cure rates of less than 30% [8], due to disease progression or to the noxious effects, both in the short and long term, associated with the intensification of cytotoxic treatments [9]. Such poor prognosis is a powerful reason for the urgent need of novel, innovative therapies that are more specific and efficient in cases of no response.

On this basis, the concept and application of personalised medicine has led to a paradigm shift in the care of children with haematological malignancies. The exploration of genomics through the advent of next-generation sequencing (NGS) has enabled the identification of oncogenic tumour biomarkers and the progressive understanding of drug resistance mechanisms acquired by blasts at relapse. This constitutes a game-changing tool to potentially optimise molecular diagnosis, identify high-risk patients, find druggable mutations and support patient follow-up through time [10].

Within the possibilities offered by NGS, targeted panels are the most frequently used technique in clinical settings, due to their highly practical nature when compared to the whole genome or exome sequencing. Nevertheless, we must bear in mind a predominance of panels designed for adult cancer in the market [11]. The optimisation of panels by independent research groups to match paediatric molecular alterations has become the ultimate diagnostic approach; this is a cost-effective and flexible technique that allows for fast incorporation of the data that arises from clinical trials and publications in a timely manner [12].

Here, we report the experience of a paediatric haemato-oncology department from a tertiary hospital in Spain, performing molecular characterisation in a cohort of paediatric patients with acute leukaemia with the objective of potentially offering them targeted therapy.

## 2. Materials and Methods

### 2.1. Patients

Written informed consent to participate, approved by the hospital’s Ethics Committee, was obtained from the parents or legal guardians of the patients.

Isolated bone marrow disease was defined as ≥25% of blast cells in bone marrow aspirate (cytomorphology) and/or as presence of pathological clonality by flow cytometry similar to diagnosis. Central nervous system (CNS) involvement was defined as the presence of >5% of nucleated cells with bast evidence in cerebrospinal fluid (CSF). Isolated extramedullary relapse was defined by imaging evidence and biopsy confirmation of infiltration. Combined relapse was defined as the presence of ≥5% of blasts (cytomorphology) or presence of pathological clonality by flow cytometry, and at least one extramedullary involvement. Refractory disease was defined as the absence of complete remission (CR) after induction treatment.

A total of 26 patients diagnosed with relapsed or refractory acute leukaemia in a tertiary hospital between 2014 and 2021 were included in this study. CNS involvement was evaluated by lumbar puncture in all patients, and other extramedullary localisations were studied according to clinical suspicion. Diagnosis was confirmed by cytomorphology and immunophenotype, following the current guidelines and World Health Organization (WHO) criteria [13]. Cytogenetic data were obtained by cellular culture and karyotype analysis. Patients were classified depending on their karyotype in hypodiploid, hyperdiploid, complex or normal. For ALL, the rearrangements in MLL and ETV6-RUNX1, E2A-PBX1 and BCR-ABL translocations were analysed by FISH. For the patients with AML, MLL rearrangements were studied.

### 2.2. Samples

Bone marrow samples were collected by bone marrow aspirate either at diagnosis and/or at the moment of relapse/refractoriness. In the cases with isolated CNS relapse, blast cells were obtained from CSF. Tumour DNA and RNA from bone marrow or CSF samples were extracted using the Chemagic Automated DNA Separation System (Chemagen^®^, Baesweiler, Germany) and RNeasy Mini Kit (Qiagen^®^, Hilden, Germany), respectively, according to the manufacturer’s instructions. DNA and RNA concentration and integrity were determined using the TECAN spectrophotometer, Qubit fluorometer, and Agilent Bioanalyzer.

### 2.3. Multiplex Ligand-Probe Amplification (MLPA)

Copy number alterations (CNAs) were analysed in B-ALL and T-ALL samples by MLPA. We used SALSA MLPA Probemix P335 ALL-IKZF1 and SALSA MLPA Probemix P383 T-ALL (MRC Holland^®^). The selection of the reference samples was performed according to the manufacturer’s instructions. In each experiment, we included at least 3 DNA samples obtained from peripheral blood of healthy individuals, with a normal copy number for the sequences detected by the target and reference probes (MRC-Holland^®^, www.mrcholland.com (accessed from July to December 2021)). CNAs were analysed using the ABI PRISM 3130 XL system and Coffalyser^®^ NET Software (v.210604.1451) MRC-Holland^®^ (Amsterdam, The Netherlands).

### 2.4. Quantitative Polymerase Chain Reaction (qPCR)

The quantification of the expression of the *CRLF2* and *WT1* genes was carried out by qPCR. The commercial TaqMan gene expression assays Hs00845692_m1 (Invitrogen^®^, Waltham, MA, USA) and the kit Ipsogen WT1 ProfileQuant (ELN) (Qiagen^®^) were used to target *CRLF2* and *WT1* genes, respectively.

*WT1* expression was detected using the Ipsogen *WT1* ProfileQuant kit following the manufacturer’s instructions. In the case of *CRLF2*, relative expression was calculated using the comparative Ct method and obtaining the relative fold-change value (2^−ΔΔCt^). At least three healthy control samples were included and a relative-fold change value in the patient sample above 2 was considered as *CRLF2* overexpression [14].

### 2.5. Next-Generation Sequencing (NGS)

DNA samples were analysed using a customised panel including two different versions. First, version 1 (v.1), and a second and updated version 2 (v.2), where 91 and 182 genes, respectively, previously related to paediatric leukaemia and sensitivity to targeted therapies, were studied (Appendix A). The sequencing and the analysis of the raw data were carried out using the Illumina platform and a tailor-made pipeline, respectively. The resulting DNA sequence reads were mapped on the human reference sequence hg19 using Burrows–Wheeler Aligner (v0.7.17) [15]. PCR duplicates were removed using Picard (v2.18.25) [16], and recalibration of the reads was performed using the Genome Analysis Toolkit (GATK v4.1.4.9) [17]. SNPs and indels were detected using Pisces (v5.2.10.49) [18], Mutect2 (GATK v4.1.4.9) [19] and Lofreq (v2.1.5) [20]. Additionally, internal tandem duplications (ITD) in *FLT3* gene were studied with the software Manta (v1.6.0) [21]. Variants were annotated in the Variant Call File (VCF) with gene names, predicted functional effect, protein positions and amino-acid changes, conservation scores, and population frequency data. The somatic variants identified were filtered and classified according to the American Association for Molecular Pathology (AMP) criteria [22] using VarSeq™ v2.2.0 (Golden Helix, Inc., Bozeman, MT, USA, www.goldenhelix.com (accessed from July to December 2021). Variant allele frequency (VAF) represents the fraction of reads containing a mutation divided by the total number of reads at a given locus and is a measure of mutational abundance. In our case, since bone marrow was used for DNA isolation and blast enrichment was not performed, the VAF refers to mutational abundance in bone marrow.

### 2.6. Molecular Tumour Board

Once the results were obtained and interpreted, the identified variants were considered for their diagnostic/prognostic and/or therapeutic potential.

A molecular tumour board (MTB) was constituted in order to create a multidisciplinary discussion platform that could optimise the decisions made in the final stage of this process. The main objective of the MTB was to facilitate discussions on the relevance and actionability of variants, accessibility and pursuit of clinical trials/drugs and treatment outcomes after prescription of targeted drugs.

This personalised medicine committee comprised the expertise of a wide range of professionals, including paediatric oncologists, pathologists, pharmacists, geneticists, bioinformatics, clinical trial experts and basic researchers. Figure 1 represents a schematisation of the work flow from the moment of sample extraction to the committee’s final decision and successive steps.

### 2.7. Statistical Analysis

For the quantitative variables, median and range were used. For the qualitative variants, absolute and relative frequencies were indicated.

## 3. Results

### 3.1. Clinical Data

Twenty-six patients were included in this study. The clinical data of the patients are summarised in Table 1. Detailed clinical data are provided in Appendix A.

The median age was 7.3 years (range 0.5–16.9) at diagnosis. The median time from diagnosis to relapse was 5 months (range 1–86). The median time from diagnosis to refractoriness was 6.8 months (range 1–19.1). A total of 65% of patients were male and the most frequent diagnosis was B-ALL (69.2%), followed by T-ALL (15.4%), AML (11.5%) and MPL (3.9%). Most patients had a normal karyotype (65.5%) and did not present CNS infiltration (88.5%). The median follow up was 49 months (range 6.1–154.6).

### 3.2. Molecular Findings

#### 3.2.1. CRLF2 and WT1 Overexpression

The overexpression of the *CRLF2* and *WT1* genes has been described in acute lymphoblastic and myeloid leukaemia, respectively [23,24], and its assessment can be used, for instance, in addition to minimal residual disease, as a monitoring tool during follow-up in order to evaluate the persistence of disease and risk of relapse [25]. For that reason, the expression of *CRLF2* was analysed by qPCR in the patients diagnosed with ALL, while *WT1* was studied in AML and MPL patients. In this way, gene overexpression was studied in 21 patients (80.8%), including 18 diagnosed with ALL, 2 with AML and 1 with MPL. It was performed in a total of 24 samples, including 6 collected at the moment of diagnosis, 3 at refractoriness and 15 at relapse. Three patients (HRL 7, HRL 14 and HRL 25) were studied at two different time-points (diagnosis and relapse). *CRLF2* and *WT1* overexpression were detected in 4 patients diagnosed with ALL (15.4%, 2 with B-ALL and 2 with T-ALL) and in 3 patients diagnosed with AML (11.5%).

#### 3.2.2. Copy Number Alterations (CNAs)

The most frequent CNAs in childhood leukaemia were evaluated by MLPA, FISH and/or NGS. Principal deletions were found in *CDKN2A* (8 patients, 30.8%), *CDKN2B* (6 patients, 23%), *IKZF1* (4 patients, 15.4%) and *PAX5* (4 patients, 15.4%). Duplications were found in 6 patients (23%). The CNAs detected are shown in Appendix A.

#### 3.2.3. Genetic Alterations Identified by NGS

NGS was performed in all patients. A total of 36 bone marrow samples were sequenced, 9 (25%) using v.1 and 27 (75%) using v.2 of the panel. Of them, 13 (36%) were collected at diagnosis, 3 (8.5%) at refractoriness and 20 (55.5%) at relapse. A total of 45 clinically relevant variants according to the AMP criteria [11] were found. Of them, 4 (8.9%) were classified as Tier I-diagnostic, 5 (11%) as Tier I-prognostic, 4 (8.9%) as Tier I-therapeutic; 12 (26.7%) as Tier II-diagnostic, 5 (11%) as Tier II-prognostic and 15 (33.3%) as Tier II-therapeutic. Tier I and Tier II variants were identified in 16 patients (61.5%). Among these 45 variants, 17 (37.8%) were found at diagnosis, 3 (6.7%) at refractoriness and 25 (55.5%) at relapse. Furthermore, 30 (66.7%) corresponded to single-nucleotide variants (SNV) and 15 (33.3%) to small insertions or deletions (indels). They were mostly missense mutations (*n* = 24; 53.3%), followed by frameshift mutations due to small indels (*n* = 12; 26.7%), non-sense mutations (*n* = 5; 11%), in-frame indels (*n* = 3; 6.7%) and splice-site variants (*n* = 1; 2.2%). The most frequently altered genes identified by NGS were *FLT3, PTPN11, WT1, KRAS* (3 patients, 12%) and *PHF6, PTEN, NRAS, NT5C2* (2 patients; 8%). They mostly fall among key signaling pathways, such as the *RAS/MAPK* pathway, *PI3K-AKT* and *JAK-STAT*. Figure 2 shows the results of the advanced molecular characterisation of all the samples.

When comparing the samples analysed by NGS at diagnosis with R/r, or in consecutive relapses (HRL 24), we found that patient HRL 7 presented the same oncogenic variants in both samples, whereas patients HRL 9, HRL10, HRL 17, HRL 24 and HRL 26 showed some differences (Appendix A). Nevertheless, in most cases, these differences were in variants with a low VAF and could be the result of differences in depth. As an exception, patient HRL 26 presented at diagnosis one mutation in *PHF6* and two different mutations in *PTEN*, with a VAF between 0.3 and 0.4. At relapse, one of the *PTEN* mutation disappeared while the other doubled its proportion, becoming predominant.

### 3.3. Directed Therapy According to Genetic Characterisation

After performing a complete molecular characterisation, clinical and biological findings were discussed in the MTB in order to select high-risk patients or patients where a specific treatment could be offered. Of 26 patients, 16 (61.5%) were found to harbour a druggable mutation. Of them, 8 (50%) patients received personalised therapy after discussion by this committee. This kind of treatment was considered when no other standardised curable approach could be offered. The therapies were given inside a clinical trial whenever possible, and if not, within a compassionate use program. In addition, HSCT was performed in 7 (87.5%) patients. A total of 4 patients who received personalised treatment (50%) achieved a good response and control of their disease. Table 2 summarises this data.

The remaining eight patients did not undergo any personalised treatment because of the preclinical status of the targeted drug (HRL 5 and HRL 26), the absence of potential clinical trial enrolment with denegation of a compassionate use program (HLR 12), adequate response to a second-line treatment after a first relapse (HRL 3 and HRL 20) or exitus before treatment could be started (HRL 1, HRL 9, and HRL 21).

## 4. Discussion

The outcomes in paediatric acute leukaemia have progressively improved over the last few decades [5,26]. However, survival rates in R/r disease are poor and constitute a group of patients that are challenging to treat [27]. Recent advances in molecular technologies, such as NGS, represent a very large promise to accelerate precision medicine for the care of children with cancer, and this is especially relevant in the case of high-risk patients. In this study, we showed that an NGS-based DNA panel, in combination with other molecular techniques, can be used to identify targetable genetic alterations in a high proportion of patients with R/r paediatric acute leukaemia. This molecular information, whenever possible, was used to select a personalised treatment that was able to control the disease in half of the cases.

We used a customised NGS panel in combination with MLPA and qPCR to detect the most frequent alterations in paediatric acute leukaemia. The most frequently altered genes in our cohort of patients (*CDKN2A*, *CDKN2B*, *IKZF1*, *PAX5*, *PTNP11*, *FLT3*, *PTEN*, *CRLF2*, *WT1*, *KRAS*, *NRAS*, *PHF6*, etc) have been previously reported in paediatric leukaemia [28,29,30,31,32] and are known to be involved in biological pathways related to leukemogenesis. We could also analyse, in some cases, samples at diagnosis and at relapse/refractoriness. Since some differences existed in the oncogenic variants identified in both samples, we highlight the importance of analysing the sample that is closest to the moment of treatment. When comparing samples from the same patient, collected at diagnosis and relapse, advanced molecular characterisation provides information about the clonal evolution of the disease, as has been previously reported [33].

Although a targeted approach has the inconvenience of being restricted to a limited number of genes, it also has some important advantages in clinical practice, such as the gain in efficiency both in terms of cost and time, and the more optimal coverage of targeted areas. In fact, our approach allowed the identification of oncogenic variants in 13 (50%) of the patients and the detection of actionable alterations in 17 (65.4%) of the patients. Unfortunately, only 8 (47%) of the patients with an actionable alteration were candidates to receive targeted treatment, in addition to hematopoietic stem cell transplantation that was performed in 7 (87.5%) of them. Among the patients who received personalised treatment, half of them achieved a good response.

In this study, the most frequently altered genes were *CDKN2A* and *CDKN2B*. Mutations in cell cycle regulators, such as deletions of *CDKN2A/B* gene (frequency: 5–20% in B-cell precursor and 60–80% in T-cell ALL), have a negative impact in the function of tumour suppressor genes such as *TP53* and *RB1* [28]. We found *CDKN2A/B* deletions in 2/4 (50%) T-ALL patients and in 6/18 (33.3%) patients with B-ALL. In addition, concomitant *CDKN2A/B* and *PAX5* alterations were found in 4/8 patients (50%), all of them diagnosed with B-ALL, and a pathogenic somatic variant in *CDKN2A* was identified in one patient diagnosed with B-ALL. *CDKN2A/B* mutations can be potentially targeted with cyclin inhibitors, such as ribociclib or palbociclib, which are beginning to be introduced into clinical trials in children (NCT03740334) [28,34]. Ribociclib has been successfully employed in multiple malignancies, not only in ALL, but also in combination with other drugs [35,36,37]. It is an important drug to be considered, especially in T-ALL [38]. Nevertheless, resistance to cyclin inhibitors has been reported in patients with some concomitant mutations, such as *RB1*, *p16* and especially *TP53* [39,40,41,42,43]. This fact was studied before offering cyclin inhibitors. In our study, one patient that harboured a *CDKN2A* deletion and another one with a loss of function mutation in *CDKN2A* were treated with ribociclib. Response was maintained only in the first patient (50%).

Probably, the most common alterations in leukaemia are those found in the *RAS/MAPK* pathway (50% of relapsed B-ALL, 15% of T-ALL and 50% of AML), such as *KRAS, NRAS* and *PTPN11* mutations [28,31]. We found 16 somatic variants (42%) in genes related to the *RAS/MAPK* pathway [44,45,46]. These variants, especially *KRAS* and *NRAS* ones, are localised in hot-spot regions. Since mutations in the *RAS/MAPK* pathway have been shown to be enriched at relapse, they are believed to constitute a factor for the development of drug resistance [28]. Although initially thought to be undruggable, there are ongoing research efforts to develop *MEK* or *PIK3* inhibitors, such as trametinib [47,48], with controversial results. In this report, *KRAS* and *NRAS* mutations have been treated with trametinib, without response and fatal disease progression.

*FLT3* mutations are frequently associated with AML in about one third of cases [49]. In this cohort, *FLT3* was altered in 2/3 AML patients (67%). *FLT3* main mutations can be classified in internal tandem duplicates (ITD), present in about 25% of patients, and somatic mutations in the tyrosine kinase domain (TKD), which occurs in approximately 5% of cases. There is a number of drugs being tested for these mutations, such as midostaurin, quizartinib or sorafenib [28]. Treatment with sunitinib (*FLT3*-TKD inhibitor) and quizartinib (*FLT3*-ITD inhibitor) was offered for compassionate use and in a clinical trial (NCT03793478), respectively, to two patients harbouring a *FLT3* mutation, achieving CR of the disease with sunitinib. The results of the patient treated under the clinical trial are not discussed in this paper, as the trial is ongoing. The patient treated with sunitinib had a medullar relapse 4 months after its discontinuation (treated for 1 year). After NGS study of the relapse, the same somatic *FLT3*-TKD mutation present at diagnosis was found, and treatment with sunitinib was restarted after a second HSCT for an expected period of two years. This is an important example about the uncertain management of this kind of treatments and the controversy about how long to maintain them. Until now, there are no standardised guidelines nor recommendations about it.

It is known that 5–7% of patients with B-ALL show an overexpression of *CRLF2* disrupting the *JAK/STAT* pathway, and this is commonly associated with activating *JAK2* somatic mutations by rearrangements [12,43,50,51,52], rendering a possible therapeutic target for ruxolitinib, a *JAK1/2* inhibitor. In this cohort, we discovered 4 cases with *CRLF2* overexpression (15.4%). Treatment with *JAK/SAT* inhibitors, such as ruxolitinib, was given in 2 B-ALL patients (50%). All patients presented a response to this treatment.

In this cohort, all patients treated with these new drugs were candidates to receive HSCT. Treatment was sometimes started before transplantation in order to help to achieve CR or after the procedure, as a pre-emptive therapy. Although the performance of HSCT could act as a confounding factor when considering the effectiveness of targeted therapy, the case of the patient who relapsed after discontinuing sunitinib reinforces our hypothesis of its utility. Two patients died before HSCT could be performed due to disease progression.

An important fact to take into account is the difficulty in obtaining new drugs that have not been approved for paediatric R/r acute leukaemia. All new therapies should ideally be administered within the context of clinical trials [53]. However, trials are not always available for patients, and several other limitations, such as eligibility criteria, must be considered [54]. Due to this limitation, its use depends, on many occasions, upon compassionate, off-label or extended use prescriptions, with the consequent safety and efficacy concerns. The future existence of robust evidence in this field relies on the strengthening and implementation of precision medicine strategies that promote oncogenic variant identification and clinical trial enrolment [55].

In addition, previously described variants associated with prognosis have been found. For example, variants related to drug resistance, such as *NT5C2* and *MSH6*, in ALL have been linked to thiopurines resistance [56,57]. In the same way, variants are also related to a worse prognosis with higher risk of refractoriness and relapse in AML, such as *WT1* [58,59,60], or with a favourable outcome related to a better treatment response, such *FBXW7* and *NOTCH1* in T-ALL [61,62,63].

It is becoming increasingly clear that advanced molecular characterisation is an important tool in paediatric oncology. It has been employed in tertiary hospitals in the past few years, not only in acute leukaemia, but in all paediatric tumours, mostly solid tumours [64]. Understanding the molecular landscape of acute leukaemia can lead us to the possibility of better risk stratification and the pathway of personalised medicine [65]. Accordingly, at the present time, the management of the paediatric oncological patient requires a multidisciplinary approach, where different and important points of view must be integrated. In this line, in our centre, a molecular tumour board has been recently created. Paediatric haemato-oncologists, oncological geneticists and pharmacists are key pieces in its integration.

This paper summarises the recent experience in the creation of a molecular leukaemia tumour board in a Spanish tertiary hospital. Although the reported data are promising, this study has points that should be improved. For example, it does not focus on the new cytogenetic alterations that have been shown to be associated with an unfavourable prognosis. In addition, the heterogeneity and small sample size, the absence of bone marrow samples at diagnosis from all the patients and the different follow-up times of this study are limitations. New studies with a larger number of patients, in which not only the advanced molecular characterisation is taken into account but also the detailed cytogenetic study, would provide more information on this type of disease.

## 5. Conclusions

Advanced molecular characterisation, especially NGS, represents a paradigm shift in oncology. A better understanding of the disease is essential in order to improve not only diagnostic strategies, but it may also be an important therapeutic tool.

In reference to paediatric R/r acute leukaemia, the information given by molecular characterisation is a very important strategy that should be implemented as part of standard clinical practice, both at diagnosis, and more importantly, in refractory or relapsed disease.

It also should be taken into account in conventional guidelines at the moment of diagnosis for risk stratification, although information on prognostic significance of some variants is still lacking.

Personalised medicine and the implementation of new therapies are nowadays the present of contemporary oncology, and represent the pathway to the achievement of the total cure.

## Figures and Tables

**Figure 1 jpm-12-00881-f001:**
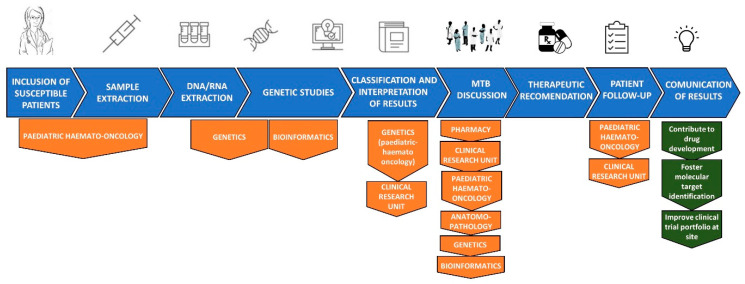
Work flow from sample collection to MTB recommendation.

**Figure 2 jpm-12-00881-f002:**
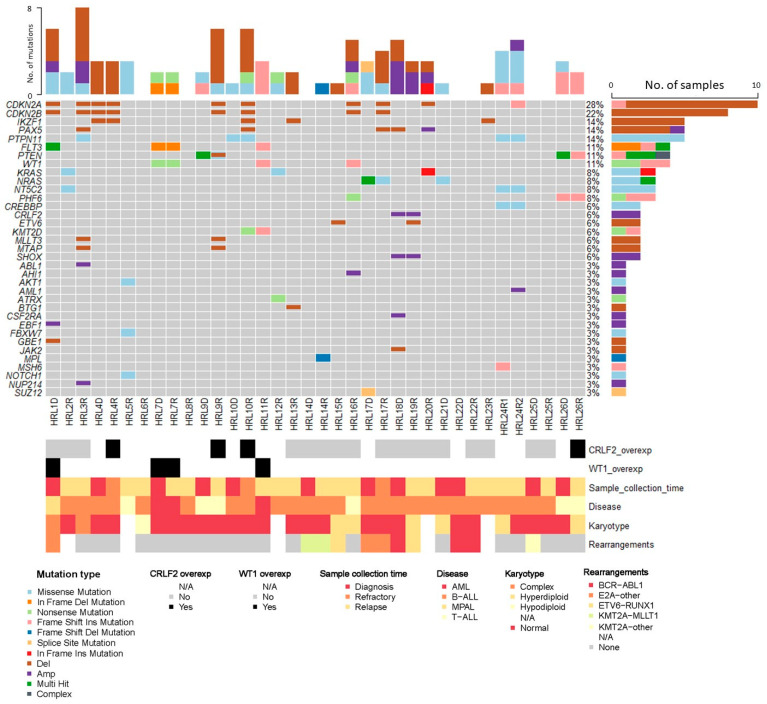
Advanced molecular characterisation of the analysed samples.

**Table 1 jpm-12-00881-t001:** Summary of clinical data.

	N (%)
SEX	
Male	17 (65.4%)
Female	9 (34.6%)
**DISEASE**	
B-ALL	18 (69.2%)
T-ALL	4 (15.4%)
AML	3 (11.5%)
MPL	1 (3.8%)
**DISEASE STATUS**	
Refractory	5 (19.2%)
Relapse:	21 (80.8%)
Isolated bone marrow	14 (66.7%)
Isolated extramedullary	3 (14.3%)
Combined relapse	4 (19%)
**KARYOTYPE (DIAGNOSIS)**	
Hypodiploid	0
Hyperdiploid	1 (3.8%)
Complex	2 (7.7%)
Normal	9 (34.6%)
**CNS INVOLVEMENT (DIAGNOSIIS)**	
Yes	3 (11.5%)
No	23 (88.5%)
**RISK GROUP (DIAGNOSIS)**	
Standard	8 (30.8%)
Intermediate	11 (42.3%)
High	7 (26.9%)
**HSCT**	
Yes	13 (50%)
No	13 (50%)
**STATUS**	
Alive	16 (61.5%)
Dead	10 (38.5%)

B-ALL: B-acute lymphoblastic leukaemia, T-ALL: T-acute lymphoblastic leukaemia, AML: acute myeloid leukaemia, MPL: mixed-phenotype leukaemia, CNS: central nervous system, HSCT: hematopoietic stem cell transplantation.

**Table 2 jpm-12-00881-t002:** Patients who received personalised treatment.

PATIENT ID	GEN	VARIANT	CLASSIFICATION (AMP)	DRUG AND DURATION (DAYS)	PRIOR/AFTER HSCT	RESPONSE
HRL 2	*NRAS*	c.183A>C(p.Gln61His)	Tier II	Trametinib (28)	After	NO
HRL 4	*CRLF2*	overexpression	Tier II	Ruxolitinib (342)	Prior	YES
HRL 7	*FLT3*	c.2503_2506delinsC (p.Asp835_Ile836delinsLeu)	Tier II	Sunitinib (383)	Prior	YES
HRL 10	*CRLF2*	overexpression	Tier II	Ruxolitinib (254)	Prior	YES
HRL 11	*FLT3*	FLT3-ITD	Tier I	Quizartinib (22)	After	N/A *
HRL 16	*CDKN2A*	deletion	Tier II	Ribociclib (108)	After	YES
HRL 17	*KRAS*	c.34G>T(p.Gly12Cys)	Tier II	Trametinib (3)	No HSCT	NO
HRL 24	*CDKN2A*	c.319dupC(p.His107fs)	Tier II	Ribociclib (67)	After	NO

* Under a Clinical trial (NCT03793478).

## Data Availability

Not applicable.

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
