# Peer review of "Advanced Molecular Characterisation in Relapsed and Refractory Paediatric Acute Leukaemia, the Key for Personalised Medicine"

_jpm, 2022, doi:10.3390/jpm12060881_

Round 1

Reviewer 1 Report

Galan-Gomez and colleagues describe data from 26 acute leukemia patients. They performed detailed molecular analysis to identify potentially targetable abnormalities. In the end, 8 patients received targeted therapy, most of them in addition to SCT, and four patients showed a clinical response.

The paper is descriptive by nature, data are generally clearly presented and the paper reads well.

Comments and suggestions:

  • NGS: a panel including 84 genes was used. It would be helpful to indicate (eg in a supplement) which 84 genes were included.
  • NGS: Please do not abbreviate BWA and AMP when used for the first time.
  • Data (in tables and in text) have too many digits, suggesting a too high level of certainty. I would suggest to use no or maximally one decimal.
  • Table 1: disease status extramedullary - does this refer to isolated extramedullary disease? And for CNS involvement: is this status at diagnosis or at relapse?  
  • Note that spaces between text and references are often missing.
  • 3.2.1.: gene expression was performed in 22 patients, 3 at refractoriness, 6 at diagnosis, and 16 at relapse; 3+6+16=25, so apparently 3 patients were analysed at two time points. Please specify this in the text.
  • Table 2: it may be helpful if it is indicated which patients also underwent SCT (and indicate whether this was prior to or after the targeted therapy).
  • 3.2.3. The sentence starting with "This mutations were... " is not clear. First, to what refers this mutations? Second, ..in 16 patient () at diagnosis (n=17; 41%)... : numbers do not match. Third, what do the percentages refer to? Percentage of what?
  • Figure 2: text is too small for easy reading and should be enlarged.
  • Figure 3: limited additional value, could be omitted.
  • Potentially targetable abnormalities were found in 17 patient, but only 8 were actually treated. It is important to provide more details about the 9 patients that were not treated, please specify for each patient the reason. Such information is crucial to understand the whole process and arguments to the final treatment decision.

Reviewer 2 Report

In the manuscript entitled „Advanced molecular characterization in relapsed and refractory paediatric acute leukaemia, the key for a personalised medicine“, by Víctor Galán-Gómez and colleagues, authors performed molecular characterization of the relapsed and refractory pediatric leukemias in order to identify potential targetable lesions and apply targeted therapy. Bearing in mind poor outcome of children with relapsed ALL, and limited improvement in the overall survival in the recent clinical trials, this study is timely and highly relevant for the field. However, although authors use broad spectrum of cytogenetic and molecular techniques to characterize these tumors, the study is very limited with the spectrum of genomic alterations that could be identified with these techniques.

Major comments:

  1. Definition of relapse and refractory disease – Authors should provide more information about criteria for diagnosing relapse and refractory disease. Currently this is not obvious from the manuscript, and to some extent even confusing to reader. For example, in the supplementary table 1, some of the patients patients who were marked as refractory (e.g., HRL 1, HRL 10), actually achieved complete remission, which contradicts the statement that they were refractory. On the other side, patients HRL 14 and HRL 22 relapsed without achieving complete remission. Finally, one of more interesting patients, HRL 7 experienced both relapse and refractory disease. Authors should make clear at which time point these events occurred (e.g., did the patient first relapse and then had a refractory disease in relapse).
  2. Identification of major cytogenetic alterations – Authors performed identification of the classical risk-stratifying cytogenetic alterations (e.g., ETV6-RUNX1, BCR-ABL1), but did not perform identification of the new cytogenetic alterations for which recent studies have shown to be associated with unfavorable prognosis (e.g., TCF-HLF or iAMP21). Although some of these alterations are not yet part of the current treatment protocols, they should be examined in the study dealing with comprehensive molecular characterization and targeted therapy, since these lesions may be driving relapse and poor response to treatment. The fact that only 11 out of 26 cases had detectable major cytogenetic alterations indicates that indeed these could have one of the new high-risk alterations.
  3. Custom NGS panel - Authors should describe the custom panel of genes used to perform NGS experiments, including the list of gene included in this panel and information whether whole gene or only particular region of interest are covered. Absence of the alteration in relapse-associated genes, e.g., CREBBP, WHSC1, NT5C2, from the list of the alterations detected in the ALL cohort (supplementary table 2) is confusing and may be explained by the custom panel used in this study.
  4. Control samples for the MLPA analysis – Authors specify that the control samples for MLPA analysis were obtained from the healthy children with “no major diseases”. Can authors explain how was major disease defined in this case? Can they ascertain that the healthy controls did not had any disease causing genetic alterations?
  5. CRLF2 and WT1 overexpression – Why authors didn’t study gene overexpression in all the cases?
  6. Comparison between diagnosis and relapse – Bearing in mind abundance of the data generated in this study it would be interesting if authors share more information about how matched diagnosis and relapse samples differ in their mutational spectrum. Did the authors observe rise of the subclone harboring pathogenic lesion from minor clone in diagnosis to major clone in relapse in any of the cases? In the current version of the manuscript this part is rather limited.
  7. In the supplementary table 2, case HRL 7 has preserved mutations in FLT3 and WT1 with allele frequencies of 35% and 42% in diagnosis to 89% and 95% in relapse, respectively. Bearing in mind normal karyotype and absence of focal deletions, how can authors explain this rise in allele frequency?

Minor comments:

  1. Line 98 – The name of the macine is Qubit
  2. Lines 125 and 126 – versions of GATK and variant callers need to be added
  3. Authors should carefully check that all the gene names are in italic

Round 2

Reviewer 1 Report

Thank you for the revision, my comments have appropriately been addressed.

Author Response

Thank you very much for your interest and your comments.

Reviewer 2 Report

Authors responded to most of my major concerns and adapted manuscript accordingly. However, after examination of the revised manuscript some of my major concerns remain, which is why I still do not consider manuscript acceptable for publication in the current form.

 Major comments:

  1. Definition of relapse and refractory disease – Authors added the missing information and adapted supplementary table 1 (in the revised manuscript supplementary table 2). However, it remains unclear how patient who achieved complete remission after induction, at the same time experienced refractory diseases after induction (e.g., case HRL1). Authors should take some time to properly scrutinize their data and further correct any other possible mistakes, and/or explain such discrepancies.
  2. Identification of major cytogenetic alterations – While I appreciate authors explanation on the objectives of the study, I do believe that detection of such rare and novel alterations may have relevance to the study and potential therapeutic significance.
  3. Custom NGS panel – Although list of genes covers majority of genes frequently found in AML and ALL, some very important relapse-associated genes in ALL are not included in the panel, e.g. WHSC1, NT5C2, which also explains their absence from the tumor samples of tested cases. Genes such as NT5C2 are particularly important for treatment adaptation because of their strong connection to resistance against thiopurines.
  4. Control samples for the MLPA analysis – Authors should make clear that the children used as healthy controls and who were tested for another reason indeed did not have any relevant genetic disease. Other reasons may also include delayed development, immunodeficiency, and these conditions may be caused by the germline alterations in the same genes found to be frequently mutated in the acute leukemias.
  5. Rise of VAF from diagnosis to relapse and definition of VAF – Variant/mutation allele frequency (VAF/MAF) estimate is generally based on the assumption of the diploid genome and 100% tumor load. That would mean that for a heterozygous mutation in all the tumor clones, VAF estimate should be around 50%. Based on explanation authors have given in their answer it appears that they did not used VAF estimates as generally accepted in the literature, but rather calculated the level of clonality, e.g., the percentage of tumor clones in which the mutation is present and not the percentage of alleles in which mutation is present. Therefore, authors should not use the term VAF as this term in the literature refers to something different. Furthermore, authors should explain in the method section that the estimated percentages given are referred to percentage of clones and not percentage of alleles. Having this in mind and the fact that authors do not provide VAF estimates, it would be necessary to indicate in the supplementary table two if the second allele for any of the mutations was affected by copy number alteration.

Author Response

Please, find attached the responding letter.
